# A Delphi process to build consensus on revised Emergency Obstetric and Newborn Care (EmONC) signal functions and levels of care

Sarah G. Moxon[1]*, Alexandra Wharton-Smith[2], Sudha Sharma[3], Jalemba Aluvaala[4], Oona M. R. Campbell[1], Gagan Gupta[5], Samantha Lobis[6], Caitlin Warthin[6], Loveday Penn-Kekana[1], Lynn P. Freedman[6] on behalf of the Delphi panel on EmONC signal functions and levels of care[¶]

1 Department of Infectious Disease Epidemiology and International Health, Faculty of Epidemiology and Population Health, London School of Hygiene & Tropical Medicine, London, United Kingdom, 2 Department of Global Health and Development, Faculty of Public Health and Policy, London School of Hygiene & Tropical Medicine, London, United Kingdom, 3 CIWEC Hospital and Travel Medicine Center, Kapurdhara Marg, Kathmandu, Nepal, 4 KEMRI-Wellcome Trust, Nairobi, Kenya, 5 UNICEF Health Section, UNICEF, New York, New York, United States of Ammerica, 6 Averting Maternal Death and Disability (AMDD), Columbia University Mailman School of Public Health, New York, New York, United States of America

¶ Membership of the Delphi panel on signal functions and levels of care is provided in the Acknowledgements (group authors).
* sarah.moxon@lshtm.ac.uk

## Abstract

The emergency obstetric care (EmOC) monitoring framework has been used for decades to monitor the availability and use of EmOC services in low- and middle-income countries (LMICs). EmOC monitoring is based around eight signal functions, a shortlist of key clinical interventions capable of averting deaths from the main direct causes of maternal mortality, categorised between two levels of care: basic and comprehensive, with a newborn resuscitation signal function added in 2009. The Re-Visioning Emergency Obstetric and Newborn Care (EmONC) Project (2020–2024) aimed to update the EmOC approach to reflect new knowledge in maternal and newborn health (MNH), and to expand the scope of the original EmOC monitoring framework. The project used technical workstreams and workshops to arrive at new proposals. This paper reports on the approach used to build consensus on a revised set of EmONC signal functions and levels of care. Using a three-round online Delphi approach, consensus (≥85%) was sought from a diverse panel of global MNH experts on EmONC signal functions and their placement at different levels of care, based on existing evidence-based guidelines. The process was iterative, each round building on the previous, and embedded in the wider Re-Visioning EmONC project; the output from each round involved coordination of inputs from multiple tiers of technical experts, including UN agencies, via technical expert groups, workstreams and workshops. The Delphi study recruited 113 experts in MNH from a range of geographic and economic settings, specialities and professions, including clinical,

**Data availability statement:** The data that support the findings of this study are openly available in LSHTM Data Compass at https://datacompass.lshtm.ac.uk/id/eprint/4762/.

**Funding:** This work was supported, in whole or in part, by the Bill & Melinda Gates Foundation [INV-001363]. Under the grant conditions of the Foundation, a Creative Commons Attribution 4.0 Generic License has already been assigned to the Author Accepted Manuscript version that might arise from this submission. The funders had no role in study design, data collection and analysis, decision to publish, or preparation of the manuscript.

**Competing interests:** The authors have declared that no competing interests exist.

academic and programme expertise. The output from the three rounds included substantial convergence, resulting in set of 25 signal functions (11 obstetric, 13 neonatal and 1 referral) that reflect the spectrum of EmONC required for women and newborns. The revised EmONC signal functions are intended as a simple approach to allow health system managers to visualise their EmONC services, and as a means to hold health systems accountable to provide the main interventions to avert preventable maternal and newborn morbidity and mortality, and stillbirths.

## Introduction

In 1997, the emergency obstetric care (EmOC) monitoring framework was published as a set of guidelines for monitoring the availability and use of emergency obstetric services in low- and middle-income countries (LMICs) [1]. This framework, with signal functions at the core, acted as a conceptually simple approach for policy makers and health sector planners, as well as researchers, to visualise national level EmOC systems and identify whether there were specific weaknesses, in order to take focused action. The original EmOC monitoring framework (1997) was built around eight signal functions, a shortlist of key clinical interventions capable of averting deaths from the main direct causes of maternal mortality that were categorised between two levels of care: basic and comprehensive (Fig 1). Surgical capacity was the key element that distinguished comprehensive level from basic. Signal functions are measured by assessing whether or not a facility performed the function in a specified timeframe (usually 3 months), which is posed as a question to a clinician in-charge at the facility. The signal functions were last revised in 2009 when one newborn signal function was added (basic neonatal resuscitation) [2] (Fig 1). Based on the signal function concept, the performance of all seven basic signal functions within a set time period defined a facility as a functioning Basic EmOC facility, while all nine signal functions were needed for a functioning Comprehensive EmOC facility. These then enabled the calculation of EmOC availability across a population. As part of the overall 2009 *Monitoring Emergency Obstetric Care (EmOC) Handbook* (also known widely as the "pink book"), EmOC availability was the first indicator in a 6-indicator set that generated information on access, utilisation and quality [2].

The common langue generated by the EmOC monitoring framework for policy makers, program planners, measurement experts, clinicians and researchers has been influential and widely used [3–7], but it has been over a decade since the signal functions and framework were revised (2009) (Fig 1). Since the 2009 revision, much has changed in the maternal and newborn health (MNH) landscape. There has been significant relative and absolute increase in health facility delivery in LMICs and a reduction in maternal deaths with greater attention towards causes of maternal morbidity and improving quality of care [8]. Newborn health is a much higher priority on the global agenda following seminal work on newborn survival [9,10] with much needed attention to neonatal deaths and stillbirths, the care needed for small and sick newborns, and improving newborn outcomes beyond survival [11,12]. The term

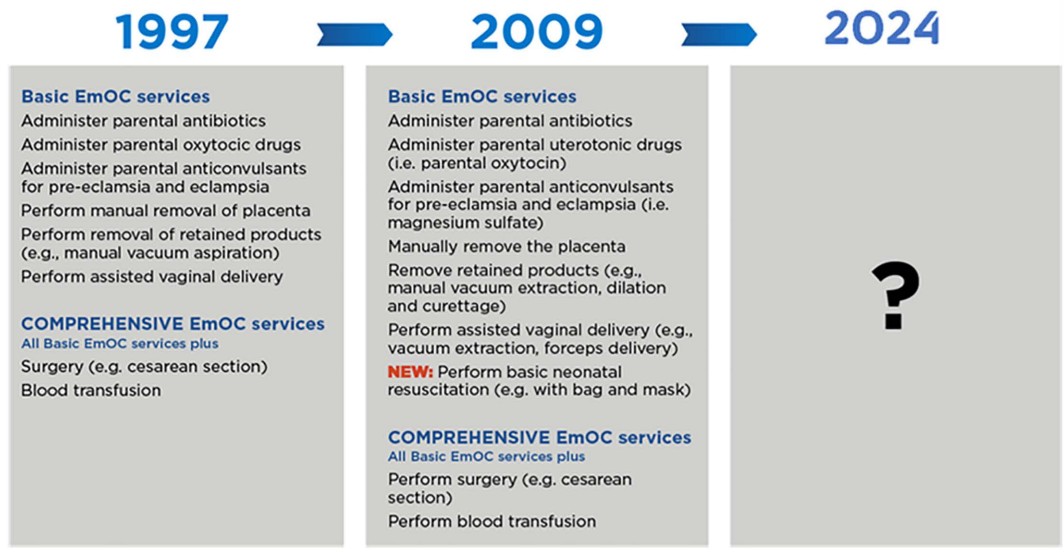

**Fig 1. Basic and Comprehensive Emergency Obstetric Care (EmOC) signal functions from 1997 to present.**

"EmONC" was adopted, with newborn advocates critiquing the EmOC monitoring approach for lacking sufficient signal functions to represent the spectrum of care required for newborns facing complications at birth [13–15].

The Re-Visioning Emergency Obstetric and Newborn Care (EmONC) Project (2020–2024) aimed to update the EmOC approach to reflect new knowledge and the desire to expand the scope of signal functions to include newborn care. The project was led by a Steering Committee coordinated by the Averting Maternal Death and Disability (AMDD) program at Columbia University Mailman School of Public Health, and included World Health Organization (WHO), UNICEF, UNFPA, and the London School of Hygiene and Tropical Medicine (LSHTM). A broad group of global stakeholder organisations and individuals formed a wider technical working group to provide substantive and strategic input throughout the process via literature reviews, four workstreams and a set of country studies.

This paper describes how multiple tiers of experts (The Re-Visioning EmONC Project Steering Committee and workstreams, individuals from UN and WHO, and a newborn expert group) input to a Delphi study, which galvanised the expertise of a wider panel of global MNH practitioners to build expert consensus on obstetric and newborn care signal functions and levels of care at which they operate, as part of the Re-Visioning EmONC project. The Delphi study facilitated a structured process to guide convergence and decision-making on revisiting the signal functions and levels of care for the revised EmONC framework.

## Methods

### Study design

The Delphi method is an approach for building consensus through eliciting and refining group judgements, especially when there is a subject of contestation or lack of empirical evidence. Developed in the 1950s, the Delphi method has since been elaborated, applied for use in health care, and for development of indicators [16,17]. The central principles of the Delphi method are the use of an expert panel, the anonymity of participants, employing multiple rounds of questionnaires to assess expert opinion and providing feedback to participants between each round [18]. Our study design was highly iterative, collecting data through three sequential rounds of questionnaires, each round building on the previous. An online Delphi was chosen as the most suitable design for gathering inputs from global experts in monitoring and measuring obstetric and newborn care.

## Ethics

Ethics approval for this study was gained from London School of Hygiene & Tropical Medicine (Ethics Reference: 26292).

## Project team

The study was coordinated by SGM with support from AW-S and involved input from the Re-Visioning EmONC Project team, Steering Committee and individuals from Workstreams 1 and 2 (Fig 2).

## Participants

As part of the planning phase, the Re-Visioning EmONC Project Steering Committee, including WHO and UN agencies, was asked to recommend a list of experts in MNH from clinical, research, programme and policymaker backgrounds. Individuals with a combination of clinical and programmatic expertise were considered ideal, although researchers and non-clinical program implementers with experience working with the EmONC signal functions were also invited. The purpose was to create a diverse sample of participants with a range of professional expertise (both obstetric and new-born; doctors and nurse-midwives) and experience in, and from, different geographies. Given the focus of the EmONC project in LMICs, participants with substantial experience working in LMICs were sought to ensure the participation of data end-users (those actually likely to use the EmONC framework for decision-making purposes), such as policy makers, programme managers and those working in planning of health services. Including both academic and professional experts was considered valuable to ensure different perspectives [19].

An invitation email and recruitment survey were prepared in English and translated into French and Spanish to promote wider participation among experts (S1 and S2 Appendix). We used convenience sampling by sharing the invitation email

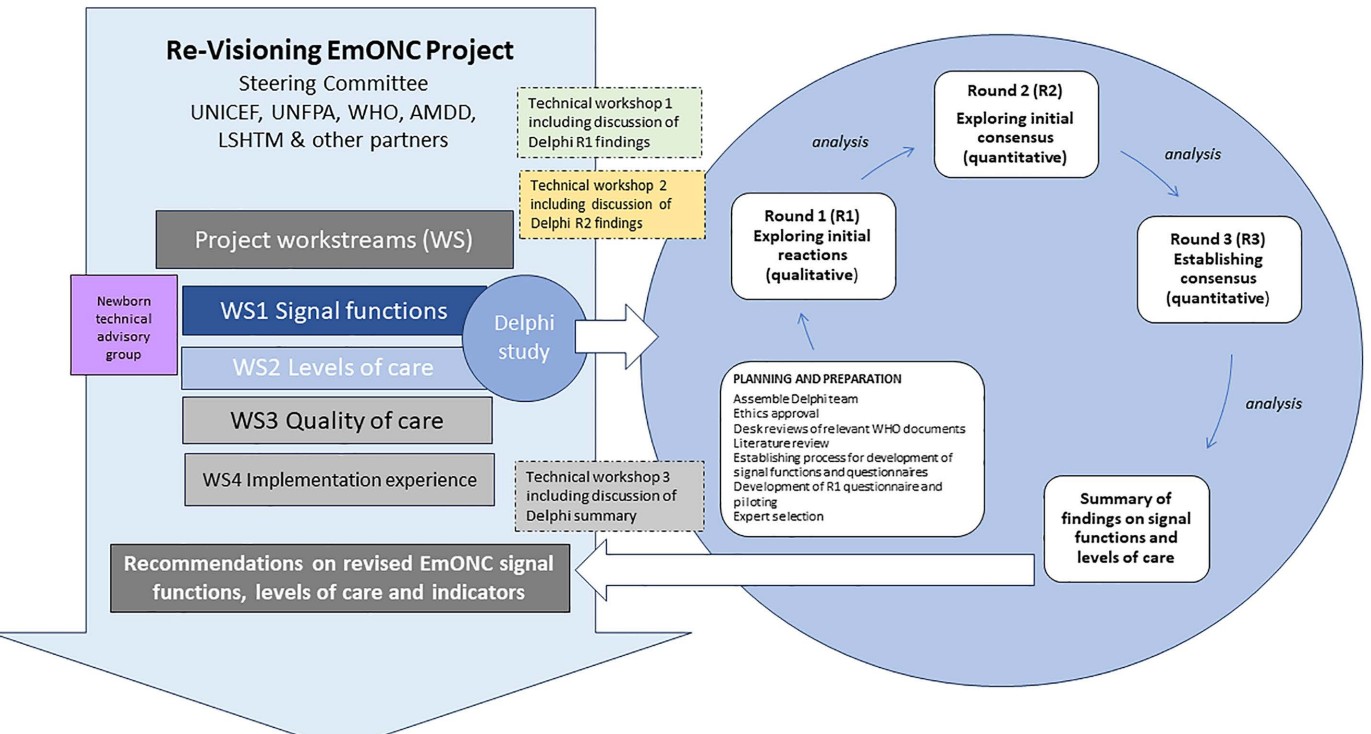

**Fig 2. Delphi study flow embedded in the wider context of the Re-Visioning EmONC Project.**

with the complete list of potential participants (generated by the Steering Committee) asking them to complete the recruitment survey, and snowball sampling by asking them to share it with their networks. The invitation email and recruitment survey were also sent to targeted project email lists, such as Every Newborn Action Plan (ENAP) and Ending Preventable Maternal Mortality (EPMM). In addition, targeted emails were drafted by Steering Committee members to regional and country-based actors (e.g., through UNFPA country teams) to encourage participation of in-country contacts and data end users. In total 279 expert participants were invited to participate in the Delphi.

**Process**

The three round online Delphi study was conducted following the schema (Fig 2), which shows how the Delphi was embedded in the wider context of the Re-Visioning EmONC Project and how inputs from the project workstreams and technical workshops fed into the different rounds of the Delphi study. A desk review of relevant clinical guidelines and standards of obstetric and newborn care was carried out to compile matrices of potential signal functions. As part of background work, a narrative literature review on signal functions was also carried out to provide context and understanding of how signal functions have been conceptualised and applied [20].

All survey rounds were piloted by the core team and other members of the Steering Committee before being refined and shared with translators. All rounds were made available in English, French and Spanish. During the process of translation, further minor amendments and refinements to wording and sentence structure were made to improve the overall clarity and translatability into French and Spanish. For each round, the French and Spanish versions were piloted by native speakers who were members of the Steering Committee or workstreams. All questionnaires were entered into and distributed by Smart Survey,© an internet based, encrypted, and secure GPDR-compliant software platform. Rounds of the Delphi were timed strategically around three project technical workshops through the lifecycle of the project. Feedback was shared between each round to members of the Delphi core group and the Steering Committee through technical workshops, which informed the content of the next round; results of the previous round were then integrated into the subsequent surveys.

The Round One (R1) questionnaire was shared with 279 maternal and newborn health experts who were invited to participate, which included informed digital written consent and participant information (S3 Appendix). The participants remained anonymous to allow them to express potentially divergent opinions freely (many participants were acquainted through professional networks, but their responses remained anonymous for the duration of the study). Following the first round, questionnaires for subsequent rounds were sent to all participants who completed the previous round. Participants were given at least two weeks to complete each round and reminders were emailed to all invitees who had not completed, or only partially completed the survey after a week. The study was carried out between November 2021 and January 2023.

**Round One (R1): Exploring expert perceptions**

R1 was open and exploratory, aiming to elicit information on what the expert participants would like to see in a revised set of signal functions and levels of care.

The R1 survey was organised into six sections and included a separate attachment with participant information (S3 Appendix):

◦ Background on project aims

◦ Digital consent

◦ Contact information

◦ Section A: Signal functions and what aspects of care signal functions should capture

- Section B: Levels of care and which criteria should influence how levels of maternal and newborn care should be organised

- Participant demographic information

For Section A and B, some questions required a quantifiable answer or a judgement between different options. All questions had free-text boxes for comments. 212 of the 279 invited participants completed Round One. Following R1 and before R2, there was a technical workshop (see Box 1).

### Round Two (R2): Exploring initial consensus – prioritising signal functions

R2 of the Delphi study focused on rating priority signal functions. The R2 survey (S4 Appendix) had four sections:

- Introduction/background refresher on aims of project

- Contact information

- Top rated Obstetric signal functions

- Top rated Newborn signal functions

Participants could choose to complete the obstetric section, the newborn section or both, depending upon their expertise. For the section on signal functions, Delphi participants were provided clear definitions of signal functions characteristics, as defined in the technical workshop (Box 1). They were then given the following scenario:

*Imagine you are working in a managerial or planning role at a district hospital in a low- or middle-income setting that you know well. The facility has a delivery caseload of at least 1000 births per year. There is a labour and delivery ward, an obstetric surgery operating room, and a neonatal unit. Think about the type of supplies, equipment, and staffing capacity typically available, as well as some of the regular challenges facing the facility in this context.*

Based on this district hospital scenario, participants were asked to rate nine priority signal functions out of 13 for obstetric and rate nine priority signal functions out of 16 for newborn care, respectively based on the four ideal characteristics/criteria for signal functions (Box 1). Nine was chosen with the aim of ending with a total of less than 20 obstetric and newborn signal functions (the 2009 monitoring framework had nine in total, but only one newborn signal function (Fig 1)). The invitation for R2 was sent to all 212 participants who completed round one.

Following R2 and before R3, there was a technical workshop (see Box 2).

### Round Three (R3): Establishing consensus – rating of signal functions and levels of care in a prototype

R3 of the Delphi study was focused on rating the signal functions and levels of care combined in a prototype (S5 Appendix). The prototype developed for R3 was derived from findings from R1 and R2 and feedback from a Re-Visioning EmONC project technical workshop, combining inputs from the workstream on levels of care [21]. R3 survey included four sections:

- Introduction/background including prototype

- Contact information

- Section A levels of care in a prototype (Fig 7)

- Section B Signal functions at each level within the prototype

In Section A on levels of care, participants were asked about their agreement with the suitability of the three levels of care presented in the prototype. In Section B on signal functions individuals were asked about their agreement with individual

signal functions (whether they agreed or disagreed with their inclusion) and their agreement with the placement of signal functions at different levels of care in the prototype. For all questions, participants who disagreed could comment on their reasons and offer alternative suggestions or views in free text.

The invitation for R3 was sent to all 131 participants who completed both R1 and R2.

Following R3, there was a technical workshop involving the Steering Committee and wider expert groups to share final outputs from the workstreams, the proposed final set of signal functions arranged by levels of care, and to receive feedback (Box 3).

## Results

In total, 113 experts participated in all three rounds of the Delphi (Fig 3). We present a summary of the demographic characteristics of this final sample (those who completed all rounds) and then we present summaries of the results of each round separately to show the process of convergence across rounds.

### Demographic data

Of the 279 experts invited to join the study, 212 participated in the first round of the Delphi (76% response rate). In R2, 131 participated (62% of those in R1) and in R3, 113 participated (85% of those in R2) (Fig 3). The final sample was comprised of a diverse and balanced group of experts in maternal and newborn health with experience working in a range of settings (both geographic and economic), specialties, and professions (Tables 1–2 and Fig 4). There was relatively equal participation throughout from those with obstetric and newborn expertise, although there was slightly higher participation on the maternal health side with good completion of both obstetric and newborn sections of the survey (Table 3). With respect to clinical expertise, 28% of respondents had a background as physicians in obstetrics and gynaecology and 23% had a clinical background in paediatrics or neonatology; a further 23% were nurses or midwives (Table 1). The final sample included a higher proportion of participants identifying as female than male (70% female to 29% male). Despite a relatively long duration of study and time between rounds, the study had a good retention rate of 53% across all three rounds overall.

### Round one results

Thematic analysis of the R1 qualitative data identified numerous sub-themes grouped in three over-arching themes: 1) a general endorsement of the signal function approach 2) a focus on emergency care; and 3) a shift away from the arrangement of signal functions in two levels (basic and comprehensive).

Here we present a selection of the relevant quantitative results alongside the thematic analysis of free text responses.

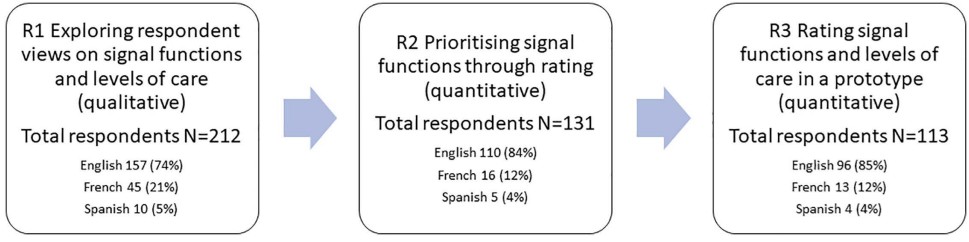

**Fig 3. Flowchart of completed Delphi rounds to build consensus on EmONC signal functions and levels of care showing survey attrition across rounds.**

**Table 1. Delphi participants' professional group(s).**

| Profession<br>*(more than one response option possible so percentages >100)* | Number of participants (N = 113) | Proportion of participants (%) |
|---|---|---|
| Obstetrician-Gynaecologist | 32 | 28 |
| Neonatologist | 13 | 12 |
| Paediatrician | 12 | 11 |
| Physician/Medical Doctor | 18 | 16 |
| Nurse | 4 | 4 |
| Neonatal Nurse | 4 | 4 |
| Midwife | 17 | 15 |
| Associate Clinician/Clinical Officer | 1 | 1 |
| Researcher/Academic in Maternal Health | 44 | 39 |
| Researcher/Academic in Newborn Health | 26 | 23 |
| Technical Advisor in Reproductive, Maternal, Newborn or Child Health | 57 | 50 |
| Programme Manager in Reproductive, Maternal, Newborn or Child Health | 27 | 24 |
| Policy-maker/Planner in Reproductive, Maternal, Newborn or Child Health | 13 | 12 |
| Government Official | 5 | 4 |
| Clinical Trainer or Instructor | 21 | 19 |
| Other | 18 | 16 |

**Table 2. Participants' experience working in different WHO regions.**

| Region<br>*(more than one option possible so percentages >100)* | Number of participants (N = 113) | Proportion of participants (%) |
|---|---|---|
| African region | 84 | 74 |
| Americas region | 26 | 23 |
| Eastern Mediterranean region | 9 | 8 |
| European region | 20 | 18 |
| South-East Asian region | 44 | 39 |
| Western Pacific region | 12 | 11 |

**Table 3. Participants who completed obstetric, newborn or both sections of survey.**

| | Number of participants (N = 113) | Proportion of participants (%) |
|---|---|---|
| Completed obstetric and newborn section | 58 | 51 |
| Completed obstetric section only | 32 | 28 |
| Completed newborn section only | 23 | 20 |

## General endorsement of signal function approach

Signal functions were described by R1 participants as *"comprehensive and concise"* as well as *"easy to report and use"*. One participant emphasised that *"they have stood the test of time"* others described this as being due to *"clarity and simplicity."*

For many participants, there was a perception that the approach should build upon past use rather than change entirely.

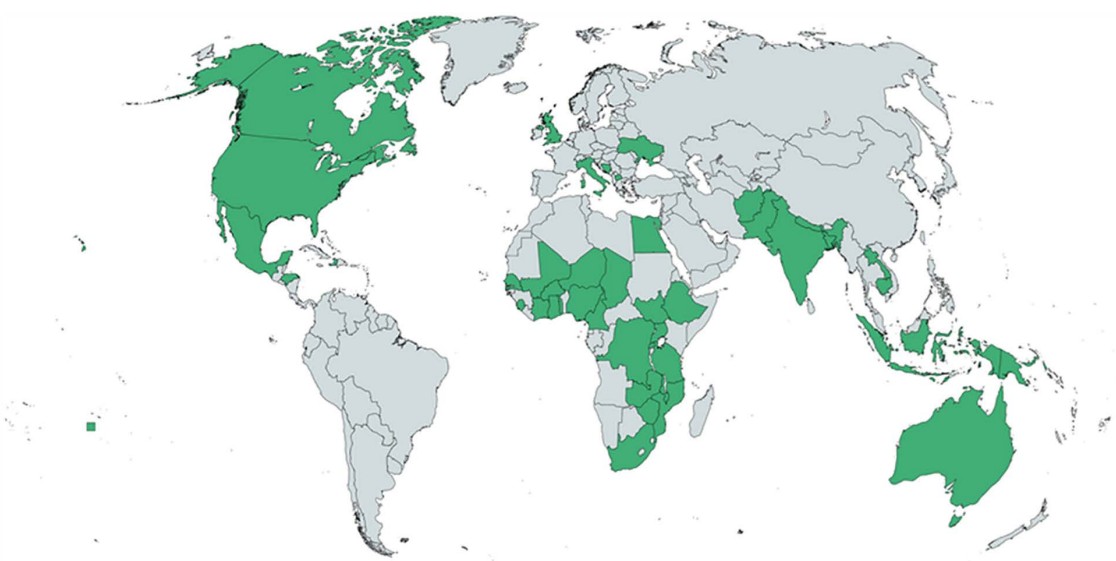

**Fig 4. Map showing countries where respondent have the most experience working in maternal and newborn health (n = 113).** Reprinted from https://mapchart.net under a CC BY 4.0, with permission from www.mapchart.net, original copyright 2014.

> *"I feel the core signal functions should remain as clear and simple as possible and that the set of indicators should continue to have a clear place (assessing a facility's performance in managing complications of mothers and newborns) in the increasingly complex landscape of health system measurement."*

Others noted that the ease of measurement and simplicity has meant that they could be integrated in routine data systems.

> *"Because they are well supported by epidemiological studies, short and explicit, it has even been possible in many countries to include the EmONC [signal functions] in the national HMIS, as in Nepal."*

For many participants the key was maintaining the approach but ensuring *"the expansion of the newborns and the explicit incorporation/mention of other emergency care for women that need to be updated".* At the same time, there was clear preference for linkages between obstetric and newborn care to ensure integration, promote togetherness of the mother baby dyad and encourage strengthening the system as a continuum.

> *"Women and their babies should be treated together in the same location, as much as possible, and health service planning should reflect that."*

Overall, these responses suggested a strong desire to maintain the concept of signal functions within the EmONC monitoring framework, but to update it to better reflect the mother-newborn dyad and ensure that with the addition of newborn signal functions, obstetric signal functions were revised accordingly.

### Focus on emergency care

An explicit objective of R1 was to explore respondents' perceptions of the focus of the revised signal functions. When asked whether signal functions should cover routine or emergency care, 92% of respondents indicated emergency care

(the original focus) and 80% of respondents indicated a desire to include routine care (including detection and prevention of complications) (Fig 5).

Within the free text qualitative responses, maintaining a focus on emergency care was framed as essential. This was seen as especially important for high mortality settings.

*"Irrespective of improving the quality of routine care, the health care systems in Low Middle Income Countries (LMICs) remain fragile, so emergency care ought to be prioritized especially improving on the emergency preparedness and complication readiness of the health facilities."*

Despite this, 80% of participants were supportive of including routine care and dimensions of care beyond clinical care alone emphasising that *"good emergency care and prevention of complications is equally important."*

Furthermore, a broadening of the dimensions of care better reflects the changing landscape of maternal and newborn health.

*"Routine care continues to be a feeder for the experience of the health system, and trust and experience with routine care influences uptake of emergency care."*

Whilst all participants articulated the importance of broader dimensions of care, especially quality, routine and respectful care, there were concerns that including routine care to the signal functions *"risks huge overlap with other assessments like SPA/SARA"* and that *"signal functions by definition shouldn't be everything because then they become the standards."* There were concerns that *"too many different kinds of things are combined into signal*

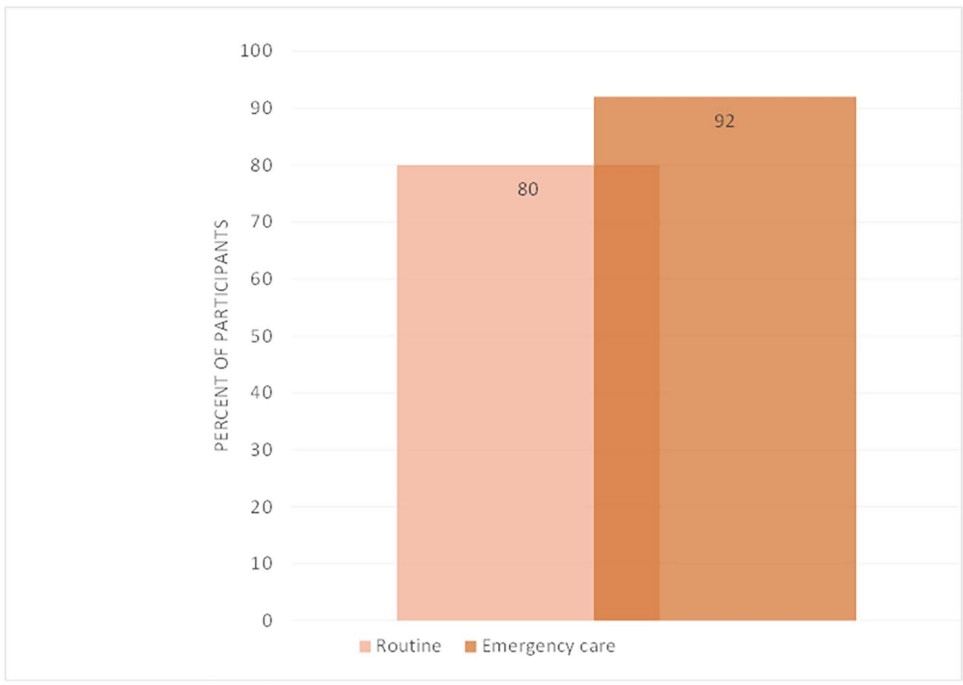

**Fig 5. Participants' views on whether emergency and routine care should be covered by the revised signal functions (n = 212) (*more than one response option possible so percentages >100).***

*functions"* suggesting routine and non-clinical care dimensions could be reflected elsewhere and might require a different measurement approach.

In summary, most participants recognised the need for the overall revision to better reflect the increased global focus on quality and respectful care, and the omission of preventive care for all mothers and newborns, but there was not clear consensus that signal functions would be the optimal way to measure these dimensions of care.

### Potential shift away from two levels of care

R1 participants were asked about the optimal number of levels of care for women and newborns with complications. Over 80% of respondents indicated more than two levels would be optimal ([Fig 6]).

Participants generally endorsed the concept of signal functions reflecting different levels of care, but some participants were concerned that "*Too many levels may affect organization, planning, and evaluation efforts".* However, they also felt there is often need for greater nuance to reflect health system complexity.

> *"I think 3 levels of care (primary, secondary/district and tertiary regional/national is a logical framework that is likely to be relevant and useful for most countries, even if an individual country has more than 3 levels of care or includes "transition" levels."*

In summary, the concept of arranging the signal functions by levels of care that are differentiated by functionality to perform certain interventions was broadly accepted by participants, but many were open to seeing a more nuanced reflection of health systems, especially when integrating newborn care.

Cutting across all three themes, participants emphasized the importance of a strong referral network and the need for increased attention to referral in the revised EmONC framework.

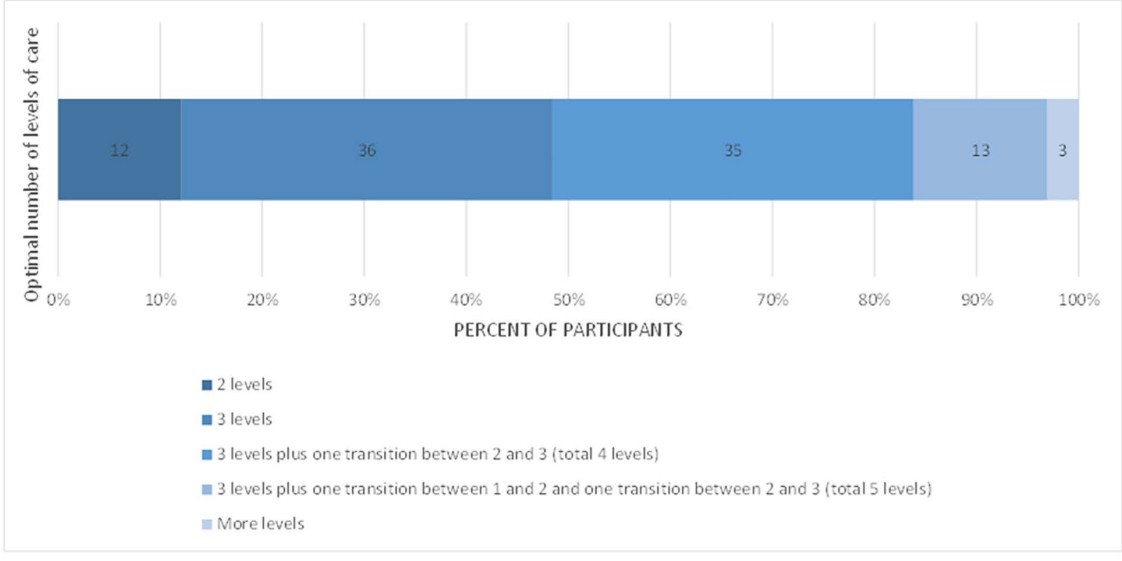

**Fig 6. Participants' views on optimal number of levels of care for women and newborns with complications (n = 212).**

**Box 1.  Technical workshop One – Characteristics of signal functions (May 2022 in Sag Harbor, NY, USA)**

Technical workshop One involved the Delphi core team to discuss the findings from R1 of the Delphi study. The output was a set of four key characteristics for signal functions and a shortlist of signal functions to be prioritised for round two of the study.

**CRITICAL:** The signal function should represent a clinical intervention or service performed by health workers to manage one or more complication(s) that would otherwise result in major cause/s of maternal or newborn death and disability, or stillbirth. Functions addressing routine care and not managing complications were excluded

**a TRACER:** Performance of the intervention or service should ideally indicate that multiple other components of treatment or aspects of care are also present (e.g., performance of surgery generally indicates the availability of anaesthesia).

**FREQUENT:** The intervention or service should need to be performed often enough to reflect ongoing performance or functionality of the facility. Rare interventions such as peripartum hysterectomy would thus be excluded

**SIMPLE:** The intervention or service need not necessarily be simple to perform, but it should be clearly and operationally defined and feasible to measure reliably across contexts.

**Round two results**

The objective of R2 of the Delphi study was to rate the top nine emergency obstetric and top nine newborn signal functions based on a set of criteria (Box 1) in order to generate a shorter list of signal functions.

To identify the top nine obstetric and newborn signal functions selected by participants in R2, the ratings given by participants were assigned an equivalent rank score, with a rank of 1 assigned a score of 1, a rank of 2 assigned a score of 2 and so on. Mean scores were calculated for each signal function separately for obstetric and newborn responses. The lowest score corresponding to the highest mean rank.

Summaries of the free text responses and the average rank scores are shown in the supporting information (S6 Appendix). The top ranked signal functions and summaries of comments on the signal functions were used as the basis for discussion with technical experts in the technical workshop at Tarrytown (Box 2) to form the signal function lists and prototype for R3 (Fig 7). Figures 10 and 11 show the nine top rated signal functions from round two for mothers and newborns alongside the results from R3.

**Box 2.  Technical workshop Two– Creating a prototype of signal functions and levels of care (September 2022 in Tarrytown, NY, USA)**

Technical Workshop Two brought together the steering committee and members of working groups. At this workshop, a "prototype" of a configuration of both obstetric and newborn signal functions and levels of care was presented to the workshop participants for feedback. This prototype was based on the highest rated signal functions from round two and inputs from the working group on levels of care. Feedback on the prototype was recorded and minor amendments were made to the prototype before finalising the prototype for sharing with Delphi participants for Round Three.

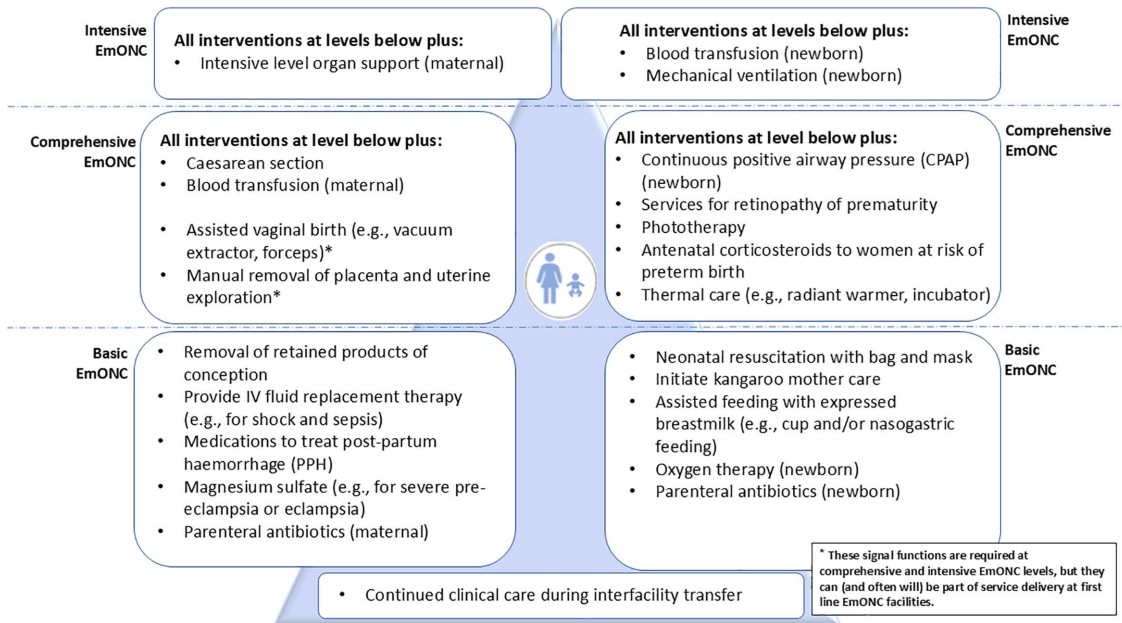

**Fig 7. Emergency obstetric and newborn care (EmONC) signal functions by three levels of care: A prototype for Round Three of the Delphi study.**

## Round three results

For R3, participants were shown a prototype of the revised EmONC signal functions arranged by levels of care (Fig 7) and were asked to rate their opinions in a series of questions on levels of care and signal functions. There were a total of 113 participants to round three, which represents a loss of 14% of participants from R2 to R3. We summarised the results in simple bar charts and tables. Free text responses were summarised (S7 Appendix) and results were compiled in a report.

Guided by the data and the overall Delphi aims, we decided to include functions which achieved a threshold of ≥85% for consensus [22]. This required ≥85% participants to agree to the inclusion of the signal function AND ≥85% of participants to agree with its placement at a specific level of care. Signal functions which fell below this threshold for either inclusion in the final set or for placement at a specific level of care were flagged as requiring further technical input and discussion.

## Section A levels of care

For the proposal to use three levels of care, 88% of participants agreed or strongly agreed (Fig 8).

## Section B Signal functions

In Section B on signal functions individuals were asked about their agreement with individual signal functions (whether they agreed or disagreed with their inclusion) and their agreement with the placement at levels of care. Fig 9 shows the percentage of participants who agreed with the placement of each individual signal functions at the different levels of care arranged in a three level configuration.

## Overall synthesis of Delphi rounds

A final report including results of all three rounds was shared with the Steering Committee and working groups prior to the third and final technical workshop.

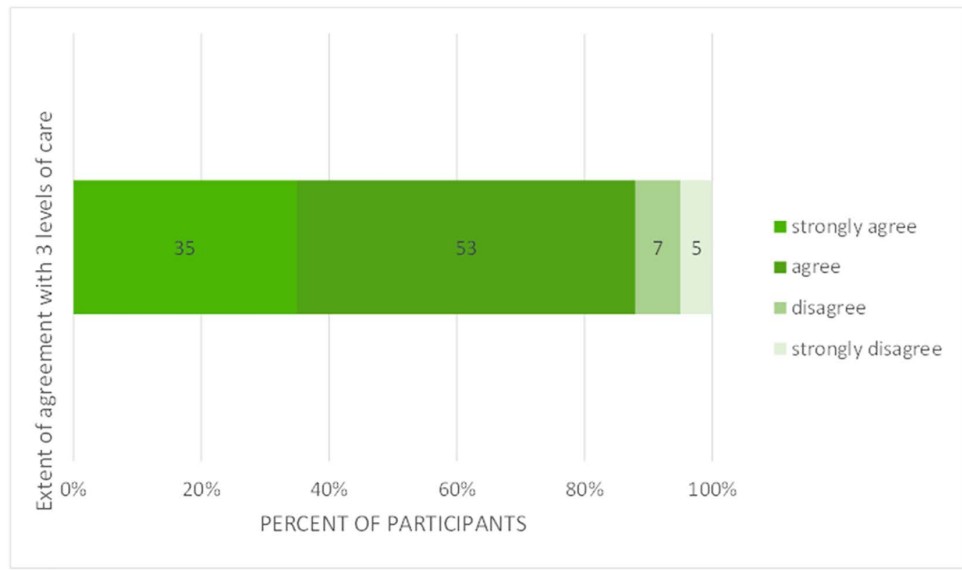

**Fig 8. Bar graph showing participants' opinion on using three levels of care as shown in the round three prototype (n = 113).**

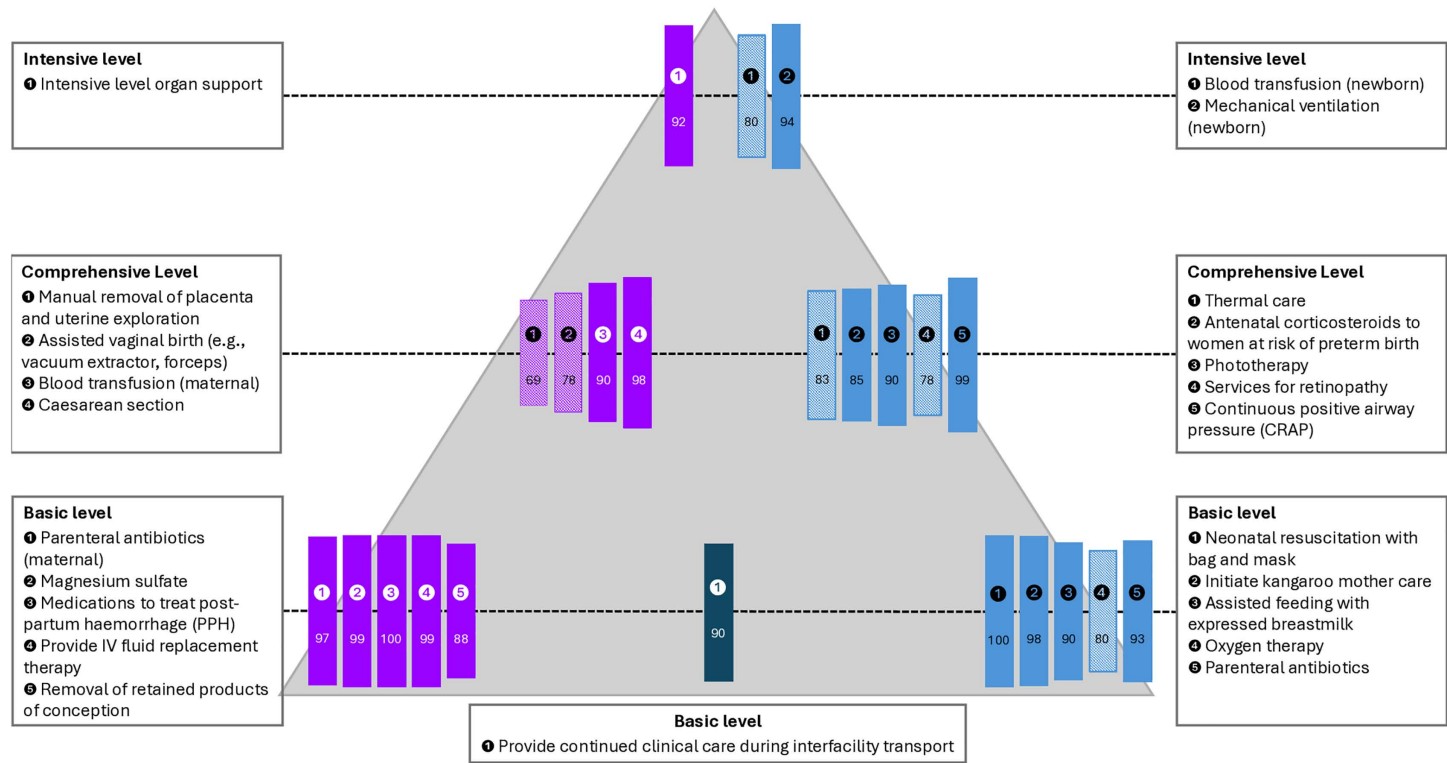

**Fig 9. Delphi participants agreement with proposed placement of signal functions at different levels of care (n = 113).**

**Box 3. Technical workshop 3– Final inputs to proposed signal functions and levels of care (November 2023, in Istanbul, Turkey)**

The third technical workshop held in November 2023 in Istanbul was the final workshop of the Re-Visioning EmONC Project. It had the overall aim of ensuring participants had a shared vision of the purpose and intended use of the revised EmONC framework. It provided an opportunity for final inputs on technical issues and reaching consensus on some of the remaining open issues with the revised framework.

Prior to the workshop, technical experts and UN agencies were consulted and amendments made to the list of signal functions and their configuration in three levels of care in order to reach a final proposed list. At the workshop, the final list of signal functions put forward as recommendations from the Re-Visioning EmONC Project were shared with the technical workshop participants.

Figs 10 and 11 show the final list of signal functions put forward as recommendations from the Re-Visioning EmONC Project following all three rounds of the Delphi and all technical workshops. The second and third column shows the status of the signal function following the second and third round and symbols indicator where further discussion with technical experts occurred, or where changes or amendments were made to the signal function following the final round of the Delphi based on inputs from the project Steering Committee, technical expert groups and/or workstreams. There are more details on wording of signal functions in the supporting information (S7 Appendix).

| Signal function | Top rated in R2 | Status following R3 | Notes on signal function |
|---|---|---|---|
| **Intensive** | | | |
| Provide intensive level organ support | | ⚡ | Viewed as aspirational for some settings, but technical discussions concluded that it was necessary for countries to reduce mortality in settings with lower mortality. |
| **Comprehensive** | | | |
| Administer antenatal corticosteroids | ✳ | ☑ ↩ | Moved from the list of newborn signal functions to the obstetric signal functions due to timing of care. Agreed at technical workshops that benefit is for the newborn, but is delivered to women at risk of imminent preterm birth as part of emergency obstetric care. |
| Perform caesarean section | ✳ | ☑ | Many respondents emphasised importance of monitoring over-use and the need for accompanying anaesthesia and pain relief. Laparotomy was also viewed as a critical surgical intervention. |
| Perform blood transfusion | ✳ | ☑ | Some respondents commented on the need for associated laboratory and storage requirements. |
| **Basic** | | | |
| Administer medications to treat PPH | ✳ | ☑ | Respondents indicated that referral is needed in severe cases for ongoing care. |
| Administer parenteral antibiotics | ✳ | ☑ | Antibiotic stewardship considered important. |
| Administer magnesium sulfate | ✳ | ☑ | Respondents indicated that referral is needed in severe cases for ongoing care. |
| Remove retained products of conception | ✳ | ☑ | Some respondents concerned that abortion and post abortion care should be more explicitly mentioned. |
| Perform manual removal of placenta | ✳ | ⚡ | Viewed as aspirational in some settings at the basic level, but technical discussions agreed to keep at basic level. |
| Perform assisted vaginal birth | | ⚡ | Long standing controversy over the placement of assisted vaginal birth at basic level, frequently not performed in some settings. Technical discussions concluded the importance of midwifery skills in performing this intervention at the basic level. |
| Provide IV fluid replacement therapy | ✳ | ✴ | Added following technical discussions in order to have a signal function for treatment of shock |
| Arrange ambulance, with trained and equipped provider, to a facility that can provide definitive care* | | ⚡ | Reflecting increased recognition that transport and referral is key to a functioning health system. Starting by focusing on emergency referral of clinically unwell mothers and babies between facilities. Aligns with WHO commitments to emergency care more generally. |

*Referral signal function to be collected with both obstetric and newborn signal functions

| | | | |
|---|---|---|---|
| Administer maternal antihypertensives | ✳ | ✂ | Removed following technical discussions, concerns over lack of clarity on situation over when it would be administered and by what route. |

**Fig 10. Emergency obstetric signal functions for the revised EmONC framework.** ⚡Lack of consensus in Delphi – resolved through discussion at technical workshops or with specific technical experts. ☑ Consensus from Delphi on signal function and level of care (more >85% agreement) ↩ Moved to different level or adapted following technical discussions. ✴Added following technical discussions. ✂Removed before R3 based on technical discussion.

| Signal function | R2 top rated | Status after R3 | Notes on signal function |
|---|---|---|---|
| **Intensive** | | | |
| Perform mechanical ventilation | | ✅ | Some respondents expressed concerns that this would be aspirational in some settings. |
| Perform screening and treatment for retinopathy of prematurity (ROP) | | ⚡ | Some respondents concerned that capacity to perform screening and treatment for ROP very dependent on the setting. Screening may be possible (and should be encouraged) at comprehensive level but treatment may only be feasible at intensive level. Technical decision to place at the intensive level. |
| **Comprehensive** | | | |
| Provide thermal care with radiant warmer or incubator | ✳ | ⚡ | Some respondents concerned this might undermine kangaroo mother care, but technical decision to include for situations when kangaroo mother care is not possible. |
| Administer oxygen therapy with pulse oximetry | ✳ | ⚡ | Lack of consensus on placing at basic or comprehensive level. Technical discussions with newborn expert group to have oxygen therapy at comprehensive level and to have oxygen therapy for stabilisation and transportation only at the basic level. |
| Provide CPAP | | ✅ | Not one of the top rated in round two but considered critical for Every Newborn Action Plan (ENAP) coverage targets and to align with WHO level 2 newborn care. |
| Provide phototherapy | ✳ | ✅ | |
| Perform blood transfusion | | ⚡ | Lack of consensus on placement at comprehensive or intensive level. Technical discussions agreed to place at comprehensive level. |
| Enable assisted feeding with expressed breastmilk | ✳ | ↩ | Moved to comprehensive level based on discussions with newborn expert group. |
| Provide IV fluids | | ✳ | Added by newborn expert group to be consistent with WHO level 2 newborn care. |
| **Basic** | | | |
| Perform newborn resuscitation with bag and mask | ✳ | ✅ | |
| Initiate and support early and exclusive breastfeeding | | ✳ | Added by newborn expert group to be consistent with WHO guidelines. |
| Administer parenteral antibiotics | ✳ | ✅ | |
| Practice immediate kangaroo mother care for preterm and LBW infants | ✳ | ✅ | |
| Administer magnesium sulfate to women at risk of preterm birth | ✳ | ✂ | Removed on advice from newborn technical expert group. |

**Fig 11. Emergency newborn signal functions for the revised EmONC framework.** ⚡Lack of consensus in Delphi – discussed at technical workshops or discussed with technical experts. ✅ Consensus from Delphi on signal function and level of care (more >85% agreement). ↩ Moved to different level or adapted following technical discussions. ✳Added following technical discussions. ✂Removed before R3 based on technical discussions.

## Discussion

We completed a three round online Delphi study with the aim of building expert consensus on EmONC signal functions and levels of care for the Re-visioning EmONC project. The study led to a substantial evolution of prioritisation of concepts, refining of signal function language and alignment between the maternal and newborn health signal functions and levels of care. The resulting set of 25 signal functions reflects the spectrum of EmONC required for both women and newborns, incorporating new obstetric signal functions and a set of signal functions for small and sick newborn care. Through its iterative and exploratory nature, the Delphi was also successful at elucidating signal functions that the previous EmONC monitoring framework did not cover, such as referral.

The first round explored participants' views on the focus of signal functions following the Re-Visioning EmONC Project. In the second round participants rated a shortlist of 28 signal functions, reducing the number of signal functions to help create a more parsimonious set. Based on collated feedback, evidence synthesis and technical input, the third round sought feedback on a prototype of signal functions and levels of care with 23 signal functions and three levels of care. Following technical discussions and a final technical workshop, the process resulted in a final set of 25 signal functions. There was consensus (88% of participants) on the decision to shift to a three-level configuration and 14/25 signal functions achieved a level of consensus (based on a threshold of ≥85%) on inclusion and placement at specific levels of care. The remaining 11 signal functions did not achieve this level of consensus, usually because of disagreements on the level of placement, or context specific implementation challenges. They were ultimately incorporated after technical expert meetings/discussions and to be consistent with WHO guidelines. Comments on the wording and language of individual signal functions was allowed through free text responses, but not the focus of each round.

The final set of signal functions includes 25 in total, 13 of which are newborn signal functions (12 new signal functions) and one is a new referral signal function (Figs 10 and 11). There are three new obstetric signal functions. This

is a significant increase from the nine signal functions used since 2009. Whilst a selection of newborn signal functions have been measured in countries previously [15,23,24], the newly proposed signal functions require rigorous field testing and careful monitoring of their implementation progress, as they are implemented in countries. The large increase in overall number of signal functions may pose measurement challenges and experience with use should be documented.

## Advantages

We adopted a highly iterative approach to the study to structure and organise the consensus process, an advantage of the Delphi method [25,26]. Through desk reviews of WHO guidelines and standards [27–30], we were able to build on existing work on newborn signal functions [14] and critiques of existing signal functions [31,32]. The Delphi study was embedded in the wider context of the Re-visioning EmONC project and was able to incorporate inputs from different workstreams as we moved across rounds [20] and adapt to the direction of the project (Fig 2).

As part of the background work for the project, we carried out a meta-narrative review on the evolution of concepts of EmONC signal functions [20]. The review identified three narrative framings – clinical, health systems and human rights – that tend to dominate the conceptualisation and use of signal functions. Potential users of EmONC monitoring framework are likely to identify with at least one of these narrative framings, with the clinical narrative often conflicting with more health systems or human rights-based narratives. Delphi studies have been widely used for health care indicators and many focus on a rating process of individual clinical indicators [16,33,34]. To avoid an exclusive focus on the clinical elements, we chose a more conceptual approach to the Delphi study. Our first round was exploratory and open. To enable participants to consider the set of signal functions for measuring EmONC from a health systems perspective, we used a scenario type question in round two and created a prototype of the EmONC signal functions and levels of care for the final round. The uniqueness of the study was in creating a structured process to guide convergence on decision making over the period of the revision while considering a wide range of differing priorities and perspectives.

Throughout the study, the Delphi was successful at elucidating conceptual issues or specific signal functions that required further technical discussion. Where consensus is difficult to reach, previous Delphi studies have used meetings to exchange views and resolve uncertainties [16], thus rounds of the Delphi were timed strategically around project technical workshops. The process of formulating specific questions for Delphi participants and communicating processes forced structured discussions to allow for reconciliation of key concepts and definitions, as well as refining of language. More details of the wording of signal functions after each round of the Delphi and the collated feedback from the Delphi rounds shared for technical discussions is available in the supporting information (S6, S7 Appendix). At the same time, the rolling deadlines of the Delphi rounds helped crystalise the thinking of the leadership from the Steering Committee and Delphi study team. There were 9 signal functions that were included in the final round for which consensus was not achieved (Fig 10 and 11). In these cases, we drew on the expertise of the Steering Committee, technical experts and workstream members (many with clinical backgrounds) to modify and refine definitions and language, and to make final judgements.

The Delphi study was successful in recruiting a diverse expert participant group from a wide range of settings, specialities and with experience working in different geographies (Table 1 and 2, Fig 4). The opportunity for participants to complete the surveys in three different languages opened participation to non-English speakers. We used repeated reminders and deadline extensions to increase the response rate where feasible and appropriate. The study had a good response and retention rate of 53%, despite the context of the pandemic/post pandemic and the relatively long study period. The size of several recent international online Delphi surveys recruiting multi-stakeholder groups ranges from 39 to 838 participants completing the final round, with retention rates of 19.5 to 87.1% [35]. The response rate demonstrated interest in the project's broader aims and commitment to the EmONC measurement approach.

## Challenges

Our study design had to incorporate multiple tiers of expert input across Delphi rounds, including the Steering Committee and workstreams (including the UN and WHO) and the newborn technical advisory group. Inevitably, decision-making was not always linear and the final set includes signal functions that were rated poorly by Delphi participants or signal functions (such as breastfeeding) which were not included in the study, but added based on strong recommendations from technical experts and key stakeholders such as WHO. This is presented transparently in the results (Figs 10 & 11). We tried to minimise these occurrences through wide sharing of the Delphi questionnaires with working groups prior to piloting and translation and having regular meetings with external technical groups (e.g., newborn advisory group). Nevertheless, group dynamics could change due to turnover of staff in different agencies and busy schedules meant not all experts could attend all meetings or review all documents by deadlines. Furthermore, the Delphi contributed recommendations, but it was the project overall that drove the final outcome.

Throughout the Delphi, there was greater consensus on obstetric signal functions than on newborn signal functions, probably because eight of these had been widely adopted and endorsed previously. By contrast, and aside from newborn resuscitation, the newborn signal functions are all new, which made discussion and final decisions more challenging. Field-testing and experience with implementation will help refine the list. Our meta-narrative review identified a strong clinical narrative across literature published on signal functions, which may partially explain the larger number of resulting newborn signal functions given their closer relationship to clinical monitoring guidelines and tools [28,29].

## Limitations

Participants working at the frontline of planning and managing EmONC facilities were some of the hardest to recruit, likely due to competing schedules and the fact that many are also providing clinical or field work in a post pandemic landscape. Our sampling relied on a convenience sampling approach, utilising mailing lists and advertising to contacts within the wider groups' existing contacts, which may have meant that some groups were overrepresented (i.e., those in known and existing networks linked to the Steering Committee). Previous studies have shown frontline providers are more likely to recommend high levels of care for more complex interventions [14]. Underrepresenting such respondents may lead to more aspirational signal functions. We tried to counterbalance this by ensuring adequate representation of experienced clinicians in technical workshops. Despite the option to complete the survey in three different languages, our attrition rates were highest in the French and Spanish participants and we had lower response rates in these languages, especially in the second round. Consequently, some expert participants who could have otherwise provided valuable input may not have been included in the final round. Our second round focused largely on what was envisioned as a Comprehensive EmONC level facility; if resources had allowed, an additional round could have looked in more detail at a Basic EmONC level facility. Whilst a Delphi study approach was seen as the most appropriate approach to achieve the specific objective of this study, such expert opinion approaches have their own limitations. Cognitive biases are a clear limitation [36]. We adopted a consensus threshold of 85% percent agreement based on and informed by analysis of findings and similar studies of this type [22,34,37]. A threshold is arbitrary and had a different threshold been selected, this could change the final recommendations.

Rather than using the Delphi as a way of forcing decisions, we used it to elucidate technical issues, guide and focus technical discussions for consensus building at workshops. Whilst every effort was made to ensure equal weight was given to all expert and technical input from multiple groups, there were inevitable areas where pragmatic judgments had to be made. We have made every effort to document all the stages of the study and the technical discussions and include the iterations of the study in the supplementary materials. Discussions that took place at technical workshops were often in the context of broader discussions on the EmONC framework (including the wider purpose and use of the framework and indicators) with access to additional discussion, research and data summaries not all of which could be shared with

Delphi participants at each round. Finally, the Delphi study could not address all the concerns or comments of the Delphi participants. The EmONC framework has a long history with multiple strands of critiques [20], not all relevant solely to the signal functions and levels of care. Wherever appropriate, relevant feedback from Delphi participants was shared with the Steering Committee and working groups to be addressed by other parts of the framework (for example, through quality and/or process indicators, or the lists of commodities).

The final diagram, (Fig 12) shows the final result of the process led by the Re-Visioning EmONC Project.

## Conclusion

We have described a Delphi process where multiple tiers of experts (The Re-Visioning EmONC Project Steering Committee and workstreams, individuals from UN and WHO, and a newborn expert group) input to a Delphi study, which served as a structured process to guide convergence and decision-making on revisiting the EmONC signal functions and levels of care for the revised EmONC framework. The resulting 25 signal functions represent the broad spectrum of EmONC required from the health system for both women and newborns, incorporating new obstetric signal functions and a set of signal functions for small and sick newborn care, as well as a new signal function for emergency referral. The revised EmONC signal functions are intended as a simple approach to allow health system managers to visualise their EmONC services within the health system, and as a means to hold systems accountable to provide the main interventions to avert preventable maternal and newborn morbidity and mortality, and stillbirths. The intention is for the signal functions to be endorsed by WHO, which will support their uptake. The challenge, in a time of resource constraints, is to advocate for their uptake and adoption by governments and organisations as they attempt to better understand their provision of life-saving healthcare services for women and newborns.

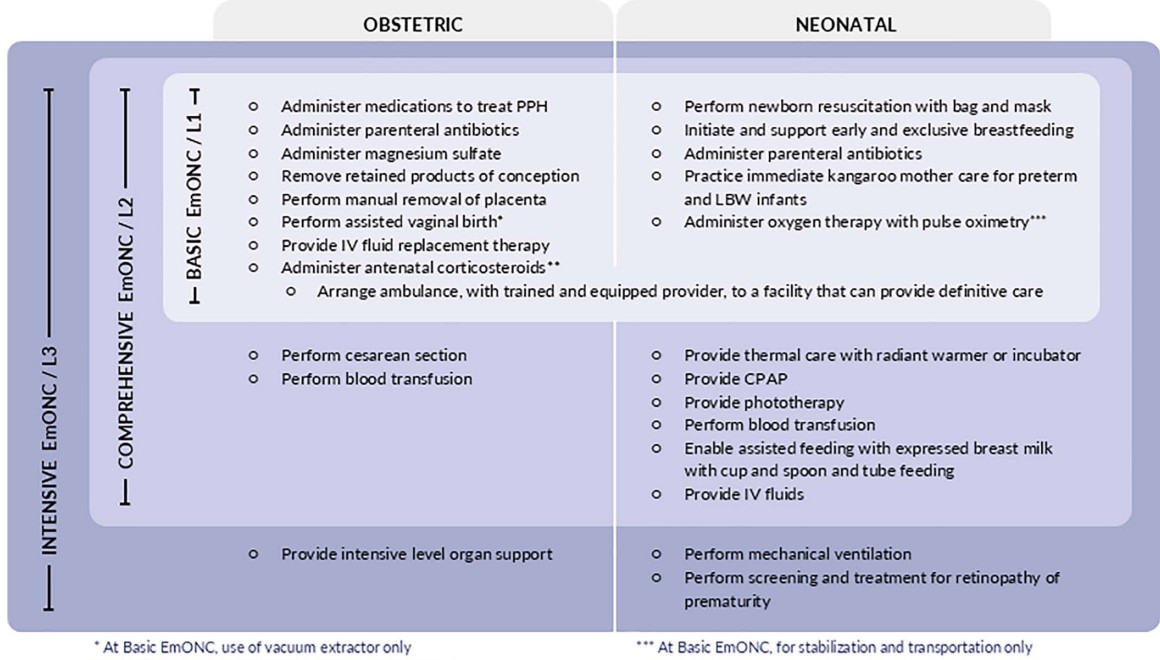

**Fig 12. Basic and comprehensive emergency obstetric and newborn care (EmONC) signal functions 2025.** Reprinted from https://emonc.org under a CC BY 4.0, with permission from Averting Maternal Death and Disability (AMDD), original copyright 2025.

## Supporting information

**S1 Appendix.  Delphi study invitation email.**
(DOCX)

**S2 Appendix.  Recruitment survey.**
(DOCX)

**S3 Appendix.  Participant information and R1 survey, including consent form.**
(DOCX)

**S4 Appendix.  R2 survey.**
(DOCX)

**S5 Appendix.  R3 survey.**
(DOCX)

**S6 Appendix.  R2 summary of results.**
(DOCX)

**S7 Appendix.  R3 free text summaries.**
(DOCX)

## Acknowledgments

We would like to thank all the membership of the Delphi panel on signal functions and levels of care (participants who completed all three survey rounds) and acknowledge their co-authorship on this manuscript: Aduragbemi Banke-Thomas (LSHTM, UK), Muna Abdullah Ali (UNFPA ESARO), Adandogou D'Almeida Adjowa Heloise (Association des Sage-Femmes, Togo), Aissata Tandina (Jhpiego, USA), Aline Mukundwa (Independent Consultant in Maternal Newborn Health), Alison Morgan (World Bank, USA), Ana Daneva-Markova (University Hospital of Gynecology and Obstetrics, Macedonia), Atnafu Getachew Asfaw (UNICEF, Uganda), Barbara Kwast (Independent Consultant in Maternal Newborn Health), Barbara Rawlins (formerly USAID, USA), Catherine Breen Kamkong (UNFPA, Bangladesh), Brenda Sequeira Dmello (University of Copenhagen, Denmark and CCBRT), Michel Brun (formerly UNFPA, New York, USA), Caroline Homer (University of Technology, Sydney and Burnet Institute, Melbourne, Australia), Caroline Williams (Alumni, LSHTM, UK), Cherrie Evans (Jhpiego, USA), Catherine Todd (FHI 360, USA), Deborah Armbruster (formerly USAID, USA), Debra Jackson (LSHTM, UK & SOPH, University of Western Cape, South Africa), Dilys Walker (Univerity of California, San Francisco, USA), Lilian Ngwongem Ngwana Ep Banmi (LSHTM, UK), Kone Seidou, Martin Dohlsten (UNICEF, Nigeria), Asim Kumar Mallick (Nil Ratan Sircar Medical College and Hospital, India), Dahada El Joud (Independent Consultant in Maternal Newborn Health), Sudha Sharma (CIWEC Hospital and Travel Medicine Center, Nepal), Emily Keyes (FHI 360, USA), Elaine Scudder (International Rescue Committee, USA), Vincenzo Pisani (Princess Christian Maternity Hospital, Sierra Leone), Emma Sacks (Johns Hopkins School of Public Health, USA), Chinyere Ezeaka (Univesity of Lagos and Lagos University Teaching Hospital, Nigeria), Francesca Tognon (Doctors with Africa, CUAMM, Italy), Fatoumata Ngayta Diop (UNFPA, Senegal), Fulgencio Sambola Estrada (FHI360, Mozambique), France Donnay (The Synergist, Brussels), Fahrija Skokić (Tuzla University Hospital Center, Bosnia and Herzegovina), Gaurav Sharma (Jhpiego, USA), T Geetha Rana (Duncan Hospital, India), Hannah Tappis (Jhpiego, USA), Harriet Ruysen (LSHTM, UK), Hema N Magge (Bill and Melinda Gates Foundation), Isabella Ochieng (Jhpiego, USA), Jos J. van Roosmalen (Leiden University Medical Centre, Netherlands), Jagjit Singh Dalal (Post Graduate Institute of Medical Sciences Rohtak Haryana, India), Jill Durocher (Gynuity Health Projects, USA), Jeanne Ann Conry (FIGO, USA), Joseph de Graft-Johnson (Save the Children, USA), Jacqueline Wallace (American Association of Birth Centers Research Committee, USA), Joleen Dako (Atrium Health Levine Children's

Hospital, Macon GA, USA), Joy E. Lawn (LSHTM, USA), Jennifer Requejo (UNICEF, USA), Judith Robb-McCord (March of Dimes, USA), Josephine Shabani (Ifakara Health Institute, Tanzania), Kara Blackburn (Burnet Institute, Australia), Kate Ramsey (Scope Impact, Finland), Carole Kenner (Council of International Neonatal Nurses and The College of New Jersey, USA), Oumou Coulibaly (University of Science and Technology and Technology of Bamako, Mali), Olena Kovalova, Katherine Semrau (Ariadne Labs, Brigham & Women's Hospital and Harvard TH Chan School of Public Health, USA), Kranti Vora (Jhpiego, USA), Etienne V. Langlois (PMNCH, WHO, Switzerland), Leonardo Contreras Osorio (Independent Consultant in Maternal Newborn Health, Nicaragua), Leah Greenspan (formerly USAID, USA), Edwin Libamba (Independent Consultant in Maternal Newborn Health, Malawi), Louise Tina Day (LSHTM, UK), Lynn P Freedman (AMDD, Columbia University, USA), Luc de Bernis (UNFPA), Luther-King Fasehun (Temple University, USA), Juan Emmanuel Dewez (LSHTM, UK), Manuela Straneo (Karolinska Institutet, Sweden), Bunsoth Mao (University of Health Sciences, Cambodia), Marcel Mbwanya (Jhpiego, USA), Marianna Zanette (WHO CC for Maternal and Child Health Institute- IRCCS Burlo Garofolo, Trieste, Italy), Zahi Lou nan Marie Laure, Elizabeth Mason (LSHTM, UK), Elahi Chowdhury (icddrb, Bangladesh), Melissa Medvedev (University of California, San Francisco, USA), Maribel Hernandez Muñoz, Mitra Kadarsih (Indonesian Midwives Association, Indonesia), Mary Kinney (University of the Western Cape, South Africa), Edgard José Narváez Delgado (UNFPA, Panama), Nathalia Makunja (Saint Francis Referral Hospital, Ifakara, Tanzania), Ngoma Kalehezo Steven, Rostand Njiki Dounou (UNFPA, Cameroon), Neena Khadka (Save the Children, USA), Narcisse Embeke (formerly USAID, Democratic Republic of Congo), Oona Campbell (LSHTM, UK), Patience Afulani (University of California, San Francisco, USA), Patricia Gomez (Jhpiego, USA), Peter Baffoe (UNICEF, Ghana), Patricia Bailey (Independent Consultant in Maternal and Newborn Health, USA), Sabrina Pestili (UNFPA, Nigeria), Ramani Ranjan, Rigoberto Castro, Robyn Churchill (formerly USAID, USA), Rebecca Penzias (LSHTM, UK), Ruth Davidge (KwaZulu-Natal Department of Health, South Africa), Sadiah Ahsan (New Beginnings Hospital, Karachi, Pakistan), Samantha Lobis (AMDD, Columbia University, USA), Sanjukta Singh (LSHTM, UK), Sarah Moxon (LSHTM, UK), Sheena M. Currie (Jhpiego, USA), Stephen Hodgins (University of Alberta, USA), Shreyashi Aryal (Kathmandu Medical College and Teaching Hospital, Nepal), Subha Sri Balakrishnan (Independent Consultant in Maternal and Newborn Health, India), Tadesse Gure (Haramaya University, College of Health and Medical science, Ethiopia), Tiyese Chimuna (UNICEF, Ethiopia), Tomomi Kitamura (UNICEF WCARO, Senegal), Theresa Shaver (formerly USAID, USA), Joseph Vyankandondera (UNFPA, Senegal), Yenealem Tadesse (Formerly JSI Research and Training Institute, Ethiopia) and Yusuf Ahmed (Levy Mwanawasa Medical University, Zambia).

Thank you to UN colleagues, including Luwei Pearson, Tedbabe Degefie Hailegebriel (UNICEF) and Michel Brun, Jean-Pierre Monet and Isabelle Moreira (UNFPA) for their contributions to technical discussions on signal functions and levels of care, and for support with initial recruitment.

We would like to acknowledge Patricia Bailey for her contributions and feedback on survey rounds and her extensive knowledge on EmOC monitoring. Thank you also to Louise Tina Day for contributions to the obstetric and newborn care signal functions and for her participation and technical support to the work on levels of care.

We would like to acknowledge the important contributions of the members of the workstream on signal functions, including Joseph Vyankandondera, Maurice Bucagu, Matthew Mathai, Subha Sri Balakrishnan, Tedbabe Degefie Hailegebriel, Hannah Blencowe for extensive discussions and review on wording and definitions of signal functions, and for piloting survey rounds.

We would like to thank all members of the newborn expert group, including Rajiv Bahl, Karen Edmonds, Ayesha de Costa, Hema Magge, Steve Wall, Gary Darmstadt, Susan Niermeyer, Joy Lawn, Lily Kak, Zulfiqar Bhutta and Luwei Pearson for their contribution to the definitions and technical content of the newborn signal functions.

We are very grateful to the translators Rokhaya Smith and Teresa Triana for their careful translations of each survey round and accompanying documents into French and Spanish.

We would also like to acknowledge and thank Maxine Pepper and Jacob Bigio who provided support with data analysis and figures, and Isabel Gouse who provided support for writing the Delphi report.

Finally, we would like to acknowledge and thank all the participants in the Delphi study, including those who did not participate in all three rounds of the study.

## Author contributions

**Conceptualization:** Sarah G. Moxon, Sudha Sharma, Jalemba Aluvaala, Oona M.R. Campbell, Loveday Penn-Kekana, Lynn P Freedman.

**Data curation:** Sarah G. Moxon, Alexandra Wharton-Smith.

**Formal analysis:** Sarah G. Moxon, Alexandra Wharton-Smith, Oona M.R. Campbell.

**Funding acquisition:** Lynn P Freedman.

**Methodology:** Sarah G. Moxon, Alexandra Wharton-Smith, Jalemba Aluvaala.

**Visualization:** Oona M.R. Campbell, Loveday Penn-Kekana, Lynn P Freedman.

**Writing – original draft:** Sarah G. Moxon, Oona M.R. Campbell, Lynn P Freedman.

**Writing – review & editing:** Sarah G. Moxon, Alexandra Wharton-Smith, Sudha Sharma, Jalemba Aluvaala, Oona M.R. Campbell, Gagan Gupta, Samantha Lobis, Caitlin Warthin, Loveday Penn-Kekana, Lynn P Freedman.

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
