## [Decision Letter · Decision Letter 0]

11 Jun 2025

PONE-D-24-58403Building consensus on revised Emergency Obstetric and Newborn Care (EmONC) signal functions and levels of care: A Delphi process and resultsPLOS ONE

Dear Dr. Moxon,

Thank you for submitting your manuscript to PLOS ONE. After careful consideration, we feel that it has merit but does not fully meet PLOS ONE’s publication criteria as it currently stands. Therefore, we invite you to submit a revised version of the manuscript that addresses the points raised during the review process.

We look forward to receiving your revised manuscript.

Kind regards,

Leonardo António Chavane, M.D., MPH, PhD

Academic Editor

PLOS ONE

Journal Requirements:When submitting your revision, we need you to address these additional requirements. 1. Please ensure that your manuscript meets PLOS ONE's style requirements, including those for file naming. The PLOS ONE style templates can be found at  https://journals.plos.org/plosone/s/file?id=wjVg/PLOSOne_formatting_sample_main_body.pdf and https://journals.plos.org/plosone/s/file?id=ba62/PLOSOne_formatting_sample_title_authors_affiliations.pdf 2. Thank you for stating the following financial disclosure:  “This work was supported, in whole or in part, by the Bill & Melinda Gates Foundation [INV-001363]. Under the grant conditions of the Foundation, a Creative Commons Attribution 4.0 Generic License has already been assigned to the Author Accepted Manuscript version that might arise from this submission.” Please state what role the funders took in the study.  If the funders had no role, please state: "The funders had no role in study design, data collection and analysis, decision to publish, or preparation of the manuscript." If this statement is not correct you must amend it as needed.  Please include this amended Role of Funder statement in your cover letter; we will change the online submission form on your behalf. 3. Please note that funding information should not appear in the Acknowledgments section or other areas of your manuscript. We will only publish funding information present in the Funding Statement section of the online submission form. Please remove any funding-related text from the manuscript.  4. We note that you have indicated that there are restrictions to data sharing for this study. For studies involving human research participant data or other sensitive data, we encourage authors to share de-identified or anonymized data. However, when data cannot be publicly shared for ethical reasons, we allow authors to make their data sets available upon request. For information on unacceptable data access restrictions, please see http://journals.plos.org/plosone/s/data-availability#loc-unacceptable-data-access-restrictions.  Before we proceed with your manuscript, please address the following prompts: a) If there are ethical or legal restrictions on sharing a de-identified data set, please explain them in detail (e.g., data contain potentially identifying or sensitive patient information, data are owned by a third-party organization, etc.) and who has imposed them (e.g., a Research Ethics Committee or Institutional Review Board, etc.). Please also provide contact information for a data access committee, ethics committee, or other institutional body to which data requests may be sent. b) If there are no restrictions, please upload the minimal anonymized data set necessary to replicate your study findings to a stable, public repository and provide us with the relevant URLs, DOIs, or accession numbers. Please see http://www.bmj.com/content/340/bmj.c181.long for guidelines on how to de-identify and prepare clinical data for publication. For a list of recommended repositories, please see https://journals.plos.org/plosone/s/recommended-repositories. You also have the option of uploading the data as Supporting Information files, but we would recommend depositing data directly to a data repository if possible. Please update your Data Availability statement in the submission form accordingly. 5. When completing the data availability statement of the submission form, you indicated that you will make your data available on acceptance. We strongly recommend all authors decide on a data sharing plan before acceptance, as the process can be lengthy and hold up publication timelines. Please note that, though access restrictions are acceptable now, your entire data will need to be made freely accessible if your manuscript is accepted for publication. This policy applies to all data except where public deposition would breach compliance with the protocol approved by your research ethics board. If you are unable to adhere to our open data policy, please kindly revise your statement to explain your reasoning and we will seek the editor's input on an exemption. Please be assured that, once you have provided your new statement, the assessment of your exemption will not hold up the peer review process.  6. One of the noted authors is a group or consortium: Delphi panel on EmONC signal functions and levels of care In addition to naming the author group, please list the individual authors and affiliations within this group in the acknowledgments section of your manuscript. Please also indicate clearly a lead author for this group along with a contact email address. 7. We note that Figure 4 in your submission contain map images which may be copyrighted. All PLOS content is published under the Creative Commons Attribution License (CC BY 4.0), which means that the manuscript, images, and Supporting Information files will be freely available online, and any third party is permitted to access, download, copy, distribute, and use these materials in any way, even commercially, with proper attribution. For these reasons, we cannot publish previously copyrighted maps or satellite images created using proprietary data, such as Google software (Google Maps, Street View, and Earth). For more information, see our copyright guidelines: http://journals.plos.org/plosone/s/licenses-and-copyright. We require you to either (1) present written permission from the copyright holder to publish these figures specifically under the CC BY 4.0 license, or (2) remove the figures from your submission: 1) You may seek permission from the original copyright holder of Figure 4 to publish the content specifically under the CC BY 4.0 license.   We recommend that you contact the original copyright holder with the Content Permission Form (http://journals.plos.org/plosone/s/file?id=7c09/content-permission-form.pdf) and the following text:“I request permission for the open-access journal PLOS ONE to publish XXX under the Creative Commons Attribution License (CCAL) CC BY 4.0 (http://creativecommons.org/licenses/by/4.0/). Please be aware that this license allows unrestricted use and distribution, even commercially, by third parties. Please reply and provide explicit written permission to publish XXX under a CC BY license and complete the attached form.” Please upload the completed Content Permission Form or other proof of granted permissions as an ""Other"" file with your submission. In the figure caption of the copyrighted figure, please include the following text: “Reprinted from [ref] under a CC BY license, with permission from [name of publisher], original copyright [original copyright year].” 2) If you are unable to obtain permission from the original copyright holder to publish these figures under the CC BY 4.0 license or if the copyright holder’s requirements are incompatible with the CC BY 4.0 license, please either i) remove the figure or ii) supply a replacement figure that complies with the CC BY 4.0 license. Please check copyright information on all replacement figures and update the figure caption with source information. If applicable, please specify in the figure caption text when a figure is similar but not identical to the original image and is therefore for illustrative purposes only. The following resources for replacing copyrighted map figures may be helpful: USGS National Map Viewer (public domain): http://viewer.nationalmap.gov/viewer/The Gateway to Astronaut Photography of Earth (public domain): http://eol.jsc.nasa.gov/sseop/clickmap/Maps at the CIA (public domain): https://www.cia.gov/library/publications/the-world-factbook/index.html and https://www.cia.gov/library/publications/cia-maps-publications/index.htmlNASA Earth Observatory (public domain): http://earthobservatory.nasa.gov/Landsat:
http://landsat.visibleearth.nasa.gov/USGS EROS (Earth Resources Observatory and Science (EROS) Center) (public domain): http://eros.usgs.gov/#Natural Earth (public domain): http://www.naturalearthdata.com/ 8. Please include captions for your Supporting Information files at the end of your manuscript, and update any in-text citations to match accordingly. Please see our Supporting Information guidelines for more information: http://journals.plos.org/plosone/s/supporting-information. 

Reviewers' comments:

Reviewer's Responses to Questions

**Comments to the Author**

1. Is the manuscript technically sound, and do the data support the conclusions?

Reviewer #1: Yes

Reviewer #2: Yes

2. Has the statistical analysis been performed appropriately and rigorously? 

Reviewer #1: Yes

Reviewer #2: N/A

3. Have the authors made all data underlying the findings in their manuscript fully available?

Reviewer #1: Yes

Reviewer #2: Yes

4. Is the manuscript presented in an intelligible fashion and written in standard English?

Reviewer #1: Yes

Reviewer #2: Yes

5. Review Comments to the Author

Reviewer #1: It is a excellent work which will help in maternal and newborn care in LMIC.

Hence I have some observation mentioned below

Title :

Building consensus on revised Emergency Obstetric and Newborn Care (EmONC) signal functions and levels of care: A Delphi process and results : As Delphi process is a method of coming to a consensus, the title can be made short and more catchy

Abstract :

Result part better to emphasized more.

Result :

Instead of terms like some/ many, these information can be expressed in percentage to quantify it

Discussion:

New signal function proposed for both maternal and newborn health better to include in discussion section for better understanding of reader.

Advantage part can be made short as there are some repeated information

Limitation also can be make specific

Reviewer #2: Thank you for the opportunity to review this paper. It represents a significant contribution to the literature, and I believe it should be published and its publication highlighted by the journal. I have some comments on the study design and conclusions and suggestions to address before publication.

The study design suited the objectives outlined by the authors. Signal functions have been in place for several decades and during that time have undergone minimal revision. Their utility has been debated and, as the overarching project suggests, re-visioning was necessary. Involving multiple stakeholders with varying expertise and experience, was also critical. While the drop out rate seems relatively high, the authors point out that a completion rate of 53% for a Dephi process such as this is not atypical. Might it have been improved and if so, how? Difficulties were faced in recruiting a suitably wide range of participants. Several groups were overrepresented, for example those who identified as academic researchers and program managers. The authors point out that participation from front line care providers was challenging. What impact might this have had on the results? In addition, while the three rounds are clearly described and the iterative process was clear, there appear to have been late additions to the list of signal functions that are not well described. Breast feeding for example, makes a final appearance on the list with the statement that it was added by the newborn expert group to be consistent with WHO guidelines. This is not addressed in the text. Did it meet criteria for a signal function or would it actually fall within routine care? There is some ambiguity about whether the signal functions cover "routine and non-clinical care dimensions." However, the paper suggests a consensus was reached on differentiating emergency from routine care. Further justification is needed for including breastfeeding as a signal function.

At the outset it was stated that in order to achieve consensus, 85% agreement would be sought. Only 14/26 or 53% actually achieved this level of consensus. Why was this the case and was it anticipated? Although it is peripherally addressed in the paper, more attention should be given to the lack of consensus and why this might be. There is mention of transparency, but what were the feelings of participants regarding the lack of consensus and was a plan in place from the beginning to address this (beyond the technical workshops)? Was that fully explained to participants?

In round 2 participants were presented with a scenario that was situated in a Comprehensive EmONC setting. Was consideration given to having participants think in a similar fashion about a first line setting, and if not, why not?

The transfer signal function, one which I believe was felt to be an important addition, is hardly discussed at all. What was the consensus around this signal function and why is it not included In figures 9, 10a or 10b?

I would like to address the signal functions themselves and the figures used to illustrate them. Figure 10a outlines the list of maternal SF’s while 10b lists the newborn SF’s There appear to be 12 in total for maternal, not including the transfer SF and 14 newborn. It is possible that I have miscounted however 10b suggests that there are 2 at the intensive level, 7 at comprehensive level and 5 at basic level (total of 14 not 13). This would be a total of 27 if one includes the referral SF. I suspect that if I am confused others will be as well! In addition it is unclear to me if the term “basic” is being used or “First level” . Figure 7 uses “first level” while figures 10 a and b use “basic”. Have we settled on that terminology?

Figure 1 is helpful historically

Figure 2 does not add to the paper. While it is important that the Delphi process be shown to be embedded in the project as a whole, the figure does not help to visualize this. What might be more helpful would be to indicate where the Delphi process and re-visioning of the signal functions is situated within the framing of the signal function (clinical, health systems and human rights).

My comment above highlights the fact that another graphic similar to Figure 7 with the updated final list would be helpful.

I would also suggest that figure 9 be revised to more clearly indicate which SF’s reached the threshold of consensus. Percentages are included but a different colour or cross hatching might help to visualize how many required more discussion and consensus building.

Finally, what are the next steps in order to operationalize these changes? This should be part of the concluding remarks. How do we develop measurement processes to track and how will countries move forward with these changes? Clearly these issues cannot be addressed fully here, but should be introduced as follow up projects.

6. PLOS authors have the option to publish the peer review history of their article (what does this mean? ). If published, this will include your full peer review and any attached files.

**Do you want your identity to be public for this peer review?** For information about this choice, including consent withdrawal, please see our Privacy Policy .

Reviewer #1: **Yes: ** Sanjoy Kumer Dey

Reviewer #2: No

---

## [Author Response · Author response to Decision Letter 1]

5 Aug 2025

Thank you for the your peer review, we have provided detailed responses in the attached table with the resubmission.

---

## [Editor Report · Decision Letter 1]

20 Aug 2025

A Delphi process to build consensus on revised emergency obstetric and newborn care (EmONC) signal functions and levels of care

PONE-D-24-58403R1

Dear Dr. Moxon,

We’re pleased to inform you that your manuscript has been judged scientifically suitable for publication and will be formally accepted for publication once it meets all outstanding technical requirements.

Kind regards,

Leonardo António Chavane, M.D., MPH, PhD

Academic Editor

PLOS ONE
---

## [Editor Report · Acceptance letter]

PONE-D-24-58403R1

PLOS ONE

Dear Dr. Moxon,

I'm pleased to inform you that your manuscript has been deemed suitable for publication in PLOS ONE. Congratulations! Your manuscript is now being handed over to our production team.

Kind regards,

on behalf of

Dr. Leonardo António Chavane

Academic Editor

PLOS ONE